# Thapsigargin and Tunicamycin Block SARS-CoV-2 Entry into Host Cells via Differential Modulation of Unfolded Protein Response (UPR), AKT Signaling, and Apoptosis

**DOI:** 10.3390/cells13090769

**Published:** 2024-04-30

**Authors:** Abeer Al Otaibi, Sindiyan Al Shaikh Mubarak, Fatimah Al Hejji, Abdulrahman Almasaud, Haya Al Jami, Jahangir Iqbal, Ali Al Qarni, Naif Khalaf Al Harbi, Ahmed Bakillah

**Affiliations:** 1King Abdullah International Medical Research Center (KAIMRC), Eastern Region, Al Ahsa 31982, Saudi Arabia; alotaibia28@ngha.med.sa (A.A.O.); alshaikhmubaraks@ngha.med.sa (S.A.S.M.); alhejjifa@ngha.med.sa (F.A.H.); iqbalja@ngha.med.sa (J.I.); qarniaa@ngha.med.sa (A.A.Q.); 2Biomedical Research Department, King Saud bin Abdulaziz University for Health Sciences (KSAU-HS), Al Ahsa 36428, Saudi Arabia; 3King Abdulaziz Hospital, Ministry of National Guard-Health Affairs (MNG-HA), Al Ahsa 36428, Saudi Arabia; 4Vaccine Development Unit, Department of Infectious Disease Research, King Abdullah International Medical Research Center, Riyadh 11481, Saudi Arabia; almasaudab@ngha.med.sa (A.A.); aljamiha@ngha.med.sa (H.A.J.); harbina2@ngha.med.sa (N.K.A.H.)

**Keywords:** apoptosis, ER stress, immune response, insulin resistance, lipid droplets, lipoproteins, SARS-CoV-2, Thapsigargin, Tunicamycin, unfolded protein response

## Abstract

Background: SARS-Co-V2 infection can induce ER stress-associated activation of unfolded protein response (UPR) in host cells, which may contribute to the pathogenesis of COVID-19. To understand the complex interplay between SARS-Co-V2 infection and UPR signaling, we examined the effects of acute pre-existing ER stress on SARS-Co-V2 infectivity. Methods: Huh-7 cells were treated with Tunicamycin (TUN) and Thapsigargin (THA) prior to SARS-CoV-2pp transduction (48 h p.i.) to induce ER stress. Pseudo-typed particles (SARS-CoV-2pp) entry into host cells was measured by Bright Glo^TM^ luciferase assay. Cell viability was assessed by cell titer Glo^®^ luminescent assay. The mRNA and protein expression was evaluated by RT-qPCR and Western Blot. Results: TUN (5 µg/mL) and THA (1 µM) efficiently inhibited the entry of SARS-CoV-2pp into host cells without any cytotoxic effect. TUN and THA’s attenuation of virus entry was associated with differential modulation of ACE2 expression. Both TUN and THA significantly reduced the expression of stress-inducible ER chaperone GRP78/BiP in transduced cells. In contrast, the IRE1-XBP1s and PERK-eIF2α-ATF4-CHOP signaling pathways were downregulated with THA treatment, but not TUN in transduced cells. Insulin-mediated glucose uptake and phosphorylation of Ser^307^ IRS-1 and downstream p-AKT were enhanced with THA in transduced cells. Furthermore, TUN and THA differentially affected lipid metabolism and apoptotic signaling pathways. Conclusions: These findings suggest that short-term pre-existing ER stress prior to virus infection induces a specific UPR response in host cells capable of counteracting stress-inducible elements signaling, thereby depriving SARS-Co-V2 of essential components for entry and replication. Pharmacological manipulation of ER stress in host cells might provide new therapeutic strategies to alleviate SARS-CoV-2 infection.

## 1. Introduction

Endoplasmic reticulum (ER) stress is a physiological condition in which the ER is overwhelmed and unable to fold and process proteins, leading to an imbalance between the production of proteins and the capacity of the ER to fold and process these proteins [1]. The disruption of ER can be caused by a range of factors, including nutrient deprivation, environmental toxins, and virus infection. When the ER is stressed, cells can activate a series of signaling pathways associated with the accumulation of misfolded proteins and the unfolded protein response (UPR), a cellular stress response pathway capable of restoring ER homeostasis by reducing protein synthesis and increasing the production of molecular chaperones to aid in enhancing protein folding capacity and degrading misfolded proteins. In case of severe or prolonged unresolved ER stress, the UPR can fail to restore homeostasis, leading to cell dysfunction and death [2,3].

Binding immunoglobulin protein (BiP) is a chaperone protein that is involved in the regulation of UPR arms and plays a central role in the maintenance of ER homeostasis by binding to unfolded proteins and preventing their aggregation [4]. The UPR is a complex signaling network that involves three main ER membrane proteins: Inositol-requiring enzyme-1 (IRE1), activating transcription factor 6 (ATF6), and protein kinase R-like ER protein kinase (PERK) [5]. When activated, these proteins initiate a series of downstream signaling events that aim to restore ER homeostasis by increasing protein folding capacity, reducing protein synthesis, and promoting the degradation of misfolded proteins. The IRE1 arm is activated in response to ER stress and leads to the splicing of *XBP1* mRNA, leading to the production of the active transcription factor XBP1, which regulates the expression of genes involved in ER stress responses and the UPR [6]. The PERK arm is activated by the phosphorylation of PERK in response to ER stress, leading to the inhibition of protein synthesis and the activation of the transcription factor ATF4, which regulates the expression of genes involved in reducing ER protein load and increasing protein folding capacity. The ATF6 arm is involved in the transport of ATF6 to the Golgi apparatus where it is cleaved and released as the active ATF6 transcription factor, leading to the regulation of genes involved in protein folding and ER stress response [7]. During a viral infection, the virus can disrupt the protein folding and processing machinery of the ER, leading to the accumulation of misfolded proteins and ER stress. This can trigger the activation of the UPR pathways as a cellular response to restore ER homeostasis. Some viruses have evolved mechanisms to exploit the cellular UPR machinery to their advantage, aiding in viral replication and spread [8]. Studies have shown that SARS-CoV-2 infection can lead to ER stress in host cells by hijacking the cellular machinery to replicate and produce viral proteins, leading to the accumulation of misfolded or unfolded proteins which trigger the UPR pathway that helps the cell cope with ER stress [9,10,11]. UPR activation during SARS-CoV-2 infection may have both protective and detrimental effects. On the one hand, the UPR can help the host cell cope with the stress induced by viral replication. On the other hand, sustained and severe UPR activation may contribute to cellular dysfunction and apoptosis. Similarly, other viruses such as human immunodeficiency virus (HIV), hepatitis C virus (HCV), influenza virus, human T-cell leukemia virus type 1 (HTLV-1), and Ebola virus have been shown to cause profound effects on cellular metabolic pathways through various mechanisms, such as UPR activation, perturbations of insulin signaling leading to alterations in glucose and lipid metabolism, and activation of pro-apoptotic signaling pathways, resulting in cell death and tissue damage [10,12,13,14,15].

ER stress can be broadly categorized into acute and chronic phases based on the duration and severity of the stress and the cellular response. Acute ER stress is typically triggered by short-term disturbances in cellular homeostasis. This can result from several factors, such as sudden increases in the demand for protein folding, viral infections, fluctuations in nutrient availability, or exposure to toxins. In general, acute ER stress is considered a protective mechanism, allowing cells to adapt to sudden changes in their environment and promoting survival. If the stress is successfully resolved, the cell returns to normal functioning. In contrast, chronic ER stress results from prolonged or unresolved disturbances in cellular homeostasis. It can be caused by persistent challenges, such as chronic inflammation, sustained elevated levels of protein synthesis, genetic mutations affecting protein folding, or conditions associated with prolonged cellular dysfunction and damage, ultimately contributing to the pathogenesis of various diseases susceptible to higher viral infection. Initially, chronic ER stress may activate the UPR, similar to acute stress but, if the stress persists and becomes severe, it can contribute to the induction of pro-apoptotic pathways leading to programmed cell death.

The severity and mortality of COVID-19 are linked to sustained ER stress-associated medical comorbidities, such as obesity, diabetes mellitus, and other chronic diseases. However, the underlying cause of increased susceptibility to viral infection in these patients’ population is not fully understood. Moreover, it remains unknown how the fate of pre-stressed infected host cells is programmed following virus infection. Here we focused our attention on the impact of acute pre-existing ER stress of two functionally independent ER stressors widely used as tools to investigate various cellular processes, including ER stress. TUN is an antibiotic that specifically inhibits the synthesis of N-linked glycoproteins leading to impairment of protein trafficking, folding, and degradation. THA, on the other hand, is a non-competitive inhibitor of the ER calcium ATPase (SERCA) pump that can disrupt calcium homeostasis, leading to UPR activation and ER stress. In addition, these two compounds have other diverse effects on cellular metabolic process beyond ER stress and UPR activation, such as modulation of autophagy, mitochondrial dysfunction and immunomodulatory effects mediating production of cytokines and chemokines [16,17,18,19].

In this study, we examined the underlying mechanisms of the effect of pre-existing acute ER stress on SARS-CoV-2 entry into host cells. Our results revealed critical control points in stressed cells that allow them to adapt and promote their survival by attenuating the virus entry and counteracting some specific stress-inducible elements signaling through modulation of UPR response and insulin/AKT signaling, offering future clues for alternate pharmacologically-based therapies targeting ER stress and UPR, which could help fight SARS-CoV-2 infection.

## 2. Materials and Methods

### 2.1. Cell Treatment and Reagents

Huh7 (hepatocyte-derived cellular carcinoma) cell lines were obtained from the Core Laboratories Facility of King Abdullah International Medical Research Center (KAIMRC) in Riyadh, Saudi Arabia. Cells were maintained in Dulbecco’s modified Eagle’s medium (DMEM) containing low glucose, 10% fetal bovine serum, L-glutamine, and 1% penicillin/streptomycin. Cells were in cultured flasks (75 cm^2^ flasks, Corning Glassworks, Corning, NY, USA) until reaching 70 to 80% confluency, then seeded into 6-well or 12-well plates for experiments. Tunicamycin (catalog #T7765) and Thapsigargin (catalog #T9033) were purchased from Sigma Aldrich (St. Louis, MO, USA). ACE2 (catalog #MBS118904) and TMPRSS2 (catalog #MABF2158) antibodies were purchased from MyBiosource (San Diego, CA, USA) and Merck Life Science UK Limited (Dorset, UK), respectively. TRIzol^™^ (catalog #15596018) was purchased from Life Technologies (Carlsbad, CA, USA). High-Capacity cDNA reverse transcription kit (catalog #4368813) was purchased from ThermoFisher Scientific (Waltham, MA, USA). SYBR^®^ Green quantitative RT-qPCR kit (catalog #10-SN10-05) was purchased from Eurogentec (San Diego, CA, USA). Primary and secondary antibodies were purchased from either Cell Signaling (Danvers, MA, USA) or Abcam (Cambridge, MA, USA). All other reagents were obtained from Fisher Scientific through a local distributor in the Kingdom of Saudi Arabia.

### 2.2. SARS-CoV-2pp Entry and Cell Viability Assays

SARS-CoV-2 pseudo-typed particles (SARS-CoV-2pp) were produced using an HIV-based lentiviral system and validated in neutralization assays as described previously [20,21,22,23]. Cells were pre-treated with ER stress inducers for 4–6 h prior to the pseudovirus transduction. At 48 h p.i., the entry of SARS-CoV-2 spike-bearing pseudoviruses into Huh7 cells was assessed using Bright Glo^TUN^ luciferase substrate (catalog #E2610; Promega, Madison, WI, USA) as previously described [20,21,22]. The relative luminescence units (RLUs) of Luc activity were measured by luminescence microplate reader (Agilent BioTek Synergy H1, Winooski, VT, USA). To determine the cell viability following treatments and pseudovirus transduction, cells were treated with Cell Titer-Glo^®^ assay reagent according to the manufacturer’s instructions (Promega, Madison, WI, USA). The obtained RLU values corresponding to the amount of ATP present in the metabolically active cells were measured by a multimode microplate reader (GloMax^®^ Explorer System, Promega, Hampshire, UK).

### 2.3. Western Blot Analysis

For the assessment of protein expression, cells were lysed with RIPA buffer and separated by SDS-PAGE 4–20% Mini-PROTEAN TGX^™^ precast protein gels (catalog #4561094; Bio-Rad, Hercules, CA, USA). The separated proteins were transferred to a PVDF membrane, blocked with TBS buffer (50 mM Tris, pH 7.6, 150 mM NaCl, 0.5% Tween-20) containing 5% BSA, and probed with different primary antibodies (1:1000 dilution) overnight at 4 °C. After several washes with TBS buffer, blots were incubated with horseradish peroxidase-conjugated secondary antibodies (1:5000 dilution) for 1 h at room temperature, washed with TBS, and developed with the Clarity Western ECL substrate (catalog #1705060; Bio-Rad, Hercules, CA, USA). The blots were then photographed, and protein bands’ densitometry was quantified with the ChemiDoc MP Imaging System from Bio-Rad (Hercules, CA, USA).

### 2.4. Immunofluorescence Staining Analysis

Huh7 cells were cultured on coverslips in 6-well plates, treated with ER stress inducers, and transduced with SARS-CoV-2pp as described above. At 48 h p.i., cells were washed with PBS and fixed with 4% formalin (15 min), permeabilized with 0.3% triton X-100 (10 min) and blocked with 3% BSA. Fixed cells were incubated overnight at 4 °C with SARS-CoV-2 Spike Protein (S1) rabbit monoclonal antibody (Cell signaling, catalog #99423; 1:250 dilution). After three washes with PBS, spike proteins were detected with Alexa Fluor 488 goat anti-rabbit antibodies (Abcam, catalog #ab 150077; 1:1000 dilution for 1h at room temperature). For F-actin detection, cells were incubated with fluorescent phalloidin solution for 30 min (Abcam catalog #ab235137). Coverslips containing the cells were mounted on glass slides in aqueous mounting media, observed and photographed using bright field and fluorescence (3D Cell Explorer, Nano Live SA, Tolochenaz, Switzerland) microscopy.

### 2.5. RNA Isolation and Quantitative Real-Time PCR (qRT-PCR)

Total RNA from treated and control cells was isolated by TRIzol^™^ as per the manufacturer’s instructions. cDNA was synthesized from total RNA with a high-capacity cDNA reverse transcription kit (ThermoFisher Scientific). Each qPCR reaction was carried out in a volume of 20 µL, consisting of 5 µL of cDNA sample (1:25 cDNA dilution) and 15 µL of SYBR Green I PCR master mix solution containing PCR buffer. The PCR was carried out by incubating the reaction mixture first for 10 min at 95 °C followed by 40 cycles of 15 s incubations at 95 °C and 1 min at 60 °C in a QuantStudio™ 6 Flex Real-Time PCR (Applied Biosystems). Data were analyzed using the ΔΔC_T_ method and presented as arbitrary units that were normalized to the expression of housekeeping genes *ACTIN* or *GAPDH*.

### 2.6. Cell Treatment and Glucose Uptake Assay

Huh7 cells were treated with 5 µg/mL of TUN or 1µM of THA for 6 h before SARS-CoV-2pp transduction (48 h p.i.). In subsequent experiments, cells were pre-treated with ER stress inducers for 4h-6h prior to SARS-CoV-2pp transduction (48 h p.i) followed by a 15 min acute insulin challenge (250 nM). Insulin-stimulated glucose uptake was determined by using a fluorescent tagged D-glucose analog (2-NBDG, 10 µM) for 1 h as described previously [24,25,26] and the resulting fluorescence was measured by Synergy H1 multimode microplate reader (BioTek, Winooski, VT, USA).

### 2.7. Lipid Droplet Staining and Lipoprotein Secretion Analysis

Huh7 cells were cultured on coverslips in 6-well plates and pre-treated with ER stress inducers for 4 h prior to SARS-CoV-2pp transduction. At 48 h p.i., media were collected, and levels of secreted lipoproteins were measured by ELISA [27]. For the lipid droplet assessment, fixed cells as described above were stained with Oil Red O to visualize lipid droplet accumulation [28]. Cells were observed and photographed using bright fields and fluorescence (3D Cell Explorer, Nano Live SA, Tolochenaz, Switzerland) microscopy.

### 2.8. Statistical Analysis

Data are presented as the mean ± SD from at least three independent experiments. Results were plotted using GraphPad Prism Software (version 5.0; GraphPad, San Di-ego, CA, USA). Statistical differences between groups were determined by a two-tailed Student *t*-test with a confidence level of 95%. A *p*-value < 0.05 was considered significant.

## 3. Results

### 3.1. Pre-Existing ER Stress Inhibits SARS-CoV-2pp Entry into Huh7 Cells Independently of ACE2 Expression

To determine whether pre-existing ER stress would affect SARS-CoV-2pp entry, we treated Huh7 cells with various concentrations of TUN (0.5–10 µg/mL) and THA (0.1–2 µM) for 6 h prior to SARS-CoV-2pp infection (35,000 RLU/mL, 48 h p.i.). Treatment with both TUN and THA significantly reduced the entry of SARS-CoV-2pp into cells in a dose-dependent manner (Figure 1A) without any cytotoxic effect, as assessed by measurement of intracellular ATP content of metabolically active cells (Figure 1B). Next, we looked at the impact of pre-existing ER stress on the expression of SARS-CoV-2 receptor ACE2 and its co-receptor serine protease TMPRSS2. TUN and THA differentially affected the levels of *ACE2* mRNA in transduced and control cells (Figure 1C). In contrast, protein levels of both ACE2 and TMPRSS2 were not significantly changed (Figure 1D). We also observed that there was no correlation between *ACE2* mRNA levels and protein expression (Figure 1C,D). The attenuation of SARS-CoV-2 entry following pre-existing ER stress-mediated by TUN and THA was confirmed by detection of immune-stained spike protein in transduced cells (Figure 1E). These results suggest that pre-existing ER stress mediating inhibition of SARCoV-2 entry was independent of cellular ACE2 expression.

### 3.2. Pre-Existing ER Stress Suppresses the Activation of UPR Gene Markers in SARS-CoV-2pp Transduced Cells

To determine whether ER stress would impact the UPR response in transduced cells, we treated Huh7 cells with 5 µg/mL of TUN and 1 µM of THA for 6 h prior to SARS-CoV-2pp infection (48 h p.i.). Strikingly, SARS-CoV-2pp (35,000 RLU/mL; 48 h p.i.) did not result in any significant UPR transcriptional changes and caused only marginal UPR activation as compared to the effects of TUN and THA (Figure 2). Treatment with TUN and THA increased mRNA levels of ER stress markers and THA appears to be a much more potent inducer (Figure 2A–D). We also observed an increase in the splicing of *XBP1* mRNA in the presence of THA but not with TUN treatment compared to vehicle-treated cells (Figure 2E). Furthermore, pre-existing ER stress resulted in significant repression of gene expression of ER stress markers (Figure 2A–D). Intriguingly, there was a poor correlation between changes in gene expression of UPR mediators and levels of protein abundance most likely due to the potential effect of translational regulators influencing the efficiency of protein biosynthesis under pre-existing ER stress conditions (Figure 2F). These results demonstrate that a pre-existing condition of ER stress prior virus infection is susceptible to counteracting the activation of unfolded protein response (UPR), thereby repressing the expression of ER stress gene markers of upstream pathways.

### 3.3. Pre-Existing ER Stress Influences Insulin Signaling Pathway and Insulin-Mediated Glucose Uptake in SARS-CoV-2pp Transduced Cells

The contribution of ER stress to the pathogenesis of insulin resistance and type 2 diabetes is well established [29,30]. Insulin signaling plays an important role in the activation of two distinct signaling cascades, PI3K/AKT (protein kinase B) and MAPK pathway [31]. Unlike TUN, THA downregulated both total protein and the phosphorylation of IRS1 and AS160 in SARS-CoV-2pp-transduced cells (Figure 3A,B). When normalization was carried out between phosphorylated and total AKT, there was an increase in the phosphorylation status in THA-transduced cells. To examine the effects of pre-existing ER stress on insulin signaling, we determined the activity of AKT and ERK as downstream targets. At the basal state, we observed that TUN and THA resulted in a significant decrease of AKT and ERK5 phosphorylation after normalization to total proteins (Figure 3A). After insulin stimulation, THA but not TUN increased AKT and ERK5 phosphorylation in transduced cells (Figure 3B). Next, we looked at the effect of pre-existing ER stress on the glucose uptake. At basal state, THA but not TUN stimulated glucose uptake in uninfected cells (Figure 3C). Pre-existing ER stress increased glucose uptake in transduced cells (Figure 3C). Acute insulin challenge induced a significant increase in glucose uptake in infected cells (Figure 3D). Furthermore, THA but not TUN further enhanced insulin-stimulated glucose uptake in transduced cells (Figure 3D). Overall, these results revealed a differential effect between TUN and THA in modulating insulin signaling and glucose uptake in transduced cells through activation of AKT/ERK phosphorylation.

### 3.4. Pre-Existing ER Stress Disrupts Lipid Droplets Formation and Lipoprotein Secretion in SARS-CoV-2pp Transduced Cells

ER stress and UPR activation can regulate cellular processes beyond ER protein folding and can play a crucial role in lipid metabolism and homeostasis [32]. Induction of ER stress by TUN and THA resulted in significant reduction of apoB and apoA-1 secretion by Huh7 cells (Figure 4A,B). In addition, SARS-CoV-2pp infection decreased the secretion of apoB and apoA1 in transduced cells (Figure 4A,B). Unlike THA, pre-existing TUN-induced ER stress enhanced the secretion of both apoB and apoA1 in transduced cells (Figure 4A,B). To further evaluate the differential effect of TUN and THA on lipid metabolism changes in transduced cells, we examined their effects on the accumulation of intracellular lipid droplets. SARSCo-V-2pp induced only a modest accumulation of intracellular lipid droplets (Figure 4C). Pre-existing ER stress resulted in less accumulation of lipid droplets in TUN transduced cells than with THA (Figure 4C). Furthermore, we noticed that cells treated with THA looked morphologically different from TUN-treated cells (Figure 4C), which may indicate a possible change in lipid composition that may have induced cell membrane morphological changes leading to alterations of droplet formation. These combined results revealed a differential effect of TUN and THA in influencing lipid droplet formation and lipoprotein secretion during pre-existing ER stress of transduced cells.

### 3.5. Pre-Existing ER Stress Influences Apoptosis Pathway in SARS-CoV-2pp Transduced Cells

Under severe ER conditions where cellular functions deteriorate, ER stress can induce cell death signaling pathways [33,34]. We found that SARS-CoV-2pp infection increased the anti-apoptotic protein BCL2 (Figure 5). To evaluate the impact of pre-existing ER stress on the apoptotic signaling pathway in transduced cells, we examined the expression of pro- and anti-apoptotic proteins involved in the activation of SAPK/JNK-induced cell death. Unlike THA, TUN induced activation of SAPK/JNK phosphorylation in transduced and control cells (Figure 5). TUN and THA resulted in blunted expression of anti-apoptotic protein BCL2 in uninfected control cells. In contrast, pre-existing ER stress resulted in a significant reduction of BCL2 expression in transduced cells (Figure 5). Furthermore, the pro-apoptotic proteins BID and BAX were unchanged. TUN had no significant changes in BAK and BAD expression in transduced cells while THA decreased the expression of BAK without affecting BAD expression in transduced cells (Figure 5). Here again, we observed a differential effect of TUN and THA in modulating apoptosis signaling, probably due to the different functional mode of actions of these two ER stress inducers. These combined data suggest that pre-existing ER stress prior to virus infection triggers cell death in transduced cells by influencing the expression of key regulator proteins in apoptosis pathways.

## 4. Discussion

Studies have shown that ER stress can affect the entry of SARS-CoV-2 into host cells by modulating the expression and function of proteins involved in viral entry [35,36,37,38]. In this study, we demonstrated that acute pre-existing ER stress prior to virus infection efficiently attenuated SARS-CoV-2 entry into Huh7 cells at various doses of TUN and THA (Figure 1). The decrease of SARS-CoV-2 entry was independent of ACE2/TMPRSS2 expression, yet TUN and THA showed differential effect on *ACE2* gene expression. The relationship between ER stress, ACE2 expression, and SARS-CoV-2 entry is complex and multifaceted. While ER stress itself may not directly influence the interaction between the viral spike protein and ACE2, it could potentially affect the expression and trafficking efficiency of ACE2 to the cell surface. Our present study did not directly address this later possibility; however, we observed a poor correlation between *ACE2* mRNA levels and cellular protein expression, which could be related to possible alterations in ACE2 trafficking and secretory pathways [39,40].

Conflicting results have been reported regarding the modulation of SARS-CoV-2 entry. A recent study reported that the activity of NUAK2, a UPR-dependent AMPK-related kinase, was able to maintain cell surface ACE2 abundance and hence promoted SARS-CoV-2 entry into cells [41]. In contrast, other studies indicated that induction of ER stress can inhibit SARS-CoV-2 entry and replication in host cells [42,43]. One plausible mechanism by which ER stress inhibits viral entry is through the upregulation of antiviral signaling pathways [44]. ER stress activates UPR-associated transcription factors, such as ATF4 and XBP1s, which can induce the expression of antiviral genes that can inhibit viral entry and replication [45,46]. In addition to the upregulation of antiviral pathways, ER stress can also directly interfere with the viral entry process by affecting the expression and localization of viral entry receptors, making it more difficult for the virus to enter host cells [47]. In the present study, infection with SARS-CoV-2 alone did not induce any significant changes in UPR at the transcriptional level, nor protein expression including the ER chaperone GRP78/BiP. This is consistent with recent studies reporting downregulation or little to no activation of the UPR arms with different MOI conditions in cell lines including human lung-derived cells [42,48]. Another study reported no activation of the ATF6 arm and only a marginal activation of IRE1α, leading to its phosphorylation and XBP1 splicing [41]. Remarkably, in the present study, acute pre-existing ER stress prior to virus infection resulted in significant repression of UPR pathways at transcriptional levels in SARS-CoV-2pp transduced cells (Figure 2). Our results are consistent with one of two conflicting reports demonstrating a reduction of cellular BiP levels in SARS-CoV-2pp transduced cells [42,49,50]. The discrepancy between our study and other reports may be due to differences in experimental settings utilizing different ER stressors during pre- or post-infection treatments. While we used a short-term THA pre-treatment (6h pre-infection) prior to infection of Huh7 cells, Shaban *et al.* adopted a long-term THA treatment (24–36 post-infection) after SARS-CoV-2 infection [42]. Interestingly, proteomics/bioinformatics analysis by Shaban et al. [42] reported that the anti-viral activity of THA in transduced cells, as demonstrated by inhibition of SARS-CoV-2 replication, was associated with significant changes of a range of metabolic pathways involving ERAD factors required for protein degradation and autophagy pathway. Clearly, in light of these findings, the precise mechanistic basis of the differential effects of TUN and THA on specific UPR pathways modulation during SARS-CoV-2 infection needs further investigation.

Accumulation of UPR in the ER can interfere with insulin signaling pathways and lead to the activation of stress kinases, such as c-Jun N-terminal kinase (JNK), which can phosphorylate insulin receptor substrate (IRS) and impair insulin receptor signaling [51,52,53,54]. It is well established that, upon insulin stimulation, AKT regulates glucose uptake in muscle, adipocytes and liver through phosphorylation, and inactivation of AS160, thereby promoting translocation of glucose transporter vesicles from intracellular stores to the plasma membrane. Furthermore, ER stress induces insulin receptor signaling through the increase in serine phosphorylation and a decrease in tyrosine phosphorylation of IRS-1, leading to insulin resistance [55]. Intriguingly, our study showed that, unlike TUN, THA downregulated the phosphorylation of IRS1 at Ser^307^ (after normalization to total IRS1) in the control cells but not in the transduced cells (Figure 3). Although the exact molecular mechanisms involved in the inhibition of total protein of IRS1 are uncertain, we may speculate that modulation of IRS1 may be seen as cellular response protection from THA-induced intracellular ER calcium disruption-promoting cell death [56]. Changes in IRS1 protein may have an impact on insulin signaling and downstream signaling pathways mediated by IRS1 [57]. While ER stress increased the phosphorylation of AKT at Ser^473^ in control cells, our study showed that TUN and THA had no significant effect on the phosphorylation of AKT in transduced cells (Figure 3), suggesting a possible contribution via other pathways modulating the AKT signaling in pre-stressed transduced cells. Our findings are consistent with a study reporting that a short period of ER stress lasting for up to 8h, in which JNKs are activated, did not inhibit insulin-stimulated AKT phosphorylation in Hepatoma cell line HepG2 [58]. In contrast, longer ER stress periods (12 h–36 h) reduced insulin-stimulated AKT phosphorylation by depleting the β chain of the mature insulin receptor at the plasma membrane of stressed cells [58].

The interactions between ER stress, insulin signaling, and glucose uptake during SARS-CoV-2 infection are complex and multifaceted. They involve the interplay of host-virus interactions, immune responses, and the overall metabolic state of the infected individual. SARS-CoV-2 infection has been associated with dysregulated glucose metabolism. Some studies have reported increased blood glucose levels in COVID-19 patients, and this hyperglycemia may have been linked to impaired insulin signaling and glucose uptake [59,60,61]. Interestingly, TUN and THA demonstrated, again, a differential effect on insulin-stimulated glucose uptake (Figure 3). THA but not TUN further enhanced the glucose uptake in control cells, suggesting a potential role of ER cytosolic calcium dysregulation in modulating glucose uptake via an independent mechanism of the insulin/AKT signaling pathway. Studies have shown that AKT substrate AS160 is implicated in both insulin and Ca^2+^-mediated glucose uptake and translocation of GLUT4 transporter protein in skeletal muscle and PBMCs [62,63]. Our study demonstrated that pre-treatment of THA and TUN prior to infection resulted in an enhanced insulin-stimulated glucose uptake, most likely through induction of GLUT translocation [64,65]. It is worth noting that the AS160 phosphorylation state alone might not be sufficient to enhance our understanding of AS160 regulation of insulin-stimulated glucose uptake in transduced cells. Therefore, the stimulatory effect of glucose uptake by THA in transduced cells may be explained as a consequence of Ca^2+^ changes inducing mobilization of GLUT2. Furthermore, a recent study provided evidence for the role of ER stress-mediated autophagic degradation of GLUT2 and GLUT4 proteins via activation of the PERK/eIF2α/ATF4 axis, which resulted in the reduction of glucose uptake in both HepG2 cells and in diabetic rats [66]. Another possible explanation is that activation of UPR during ER stress can also influence glucose metabolism by affecting the expression and activity of key enzymes involved in glucose uptake and utilization [66,67,68]. This latter possibility was not addressed in this study and is subject to future investigation.

Viral infections, in general, can lead to alterations in host cell metabolism, and this includes changes in lipid metabolism. Lipids play a crucial role in various stages of the viral life cycle, including entry, replication, and assembly [69]. The exact mechanisms by which SARS-CoV-2 might alter hepatic lipid formation are not fully understood and may involve complex interactions between the virus and host cellular pathways. Some studies have suggested that viral infections, including SARS-CoV-2, may influence lipid and lipoprotein metabolism. Changes in lipoprotein profiles have been reported in COVID-19 patients [70,71]. Most of the studies mainly focused on the impact of ER stress on patients to predict the risk of infection severity and mortality. However, factors such as inflammation, cytokine release, lipid-lowering drugs use, and systemic effects of the infection could contribute to alterations in lipid metabolism [72,73]. Intriguingly, our present study indicated that SARS-CoV-2 infection enhanced the secretion of both apoB and apoA1 in transduced cells, and this increase was significantly attenuated by pre-existing THA induced ER stress (Figure 4). Data for lipid and lipoprotein profiling in Covid-19 patients were conflicting. Alterations in lipid profiles have been reported due to SARS-CoV-2 infection, with a decline in serum total cholesterol, LDL, and HDL concentrations [74]. In contrast, another study indicated that COVID-19 patients’ liver damage was associated with a significant elevation of serum triglyceride-rich lipoproteins VLDL, IDL, LDL, and HDL and total apoB-containing VLDL and IDL subclasses, most likely due to increased expression and activity of critical enzymes involved in lipid and lipoprotein biosynthesis [75]. One of the interesting observations to emerge from the present study is that TUN and THA differentially stimulate apoA1 secretion via possible modulation of IRS1/AKT phosphorylation, resulting in improvement of glucose uptake (Figure 3 and Figure 4), which is consistent with previous reports showing that apoA1 increases glucose uptake in peripheral tissues [76,77,78,79]. Viruses, including SARS-CoV-2, can exploit lipid droplets by changing their dynamics and composition offering a source of energy and essential membrane components for viral assembly and replication [80,81]. Cholesterol esterification and triglyceride synthesis leads to the biogenesis of lipid droplets that can act as buffers that sequester misfolded proteins, free Ca^2+^, and excess lipids to alleviate ER stress [82]. As expected, because SARS-CoV-2pp can only replicate for a single round, we observed only a modest increase in intracellular lipid droplet accumulation in transduced cells compared to control cells (Figure 4). Additionally, ER stress inducers were not able to induce any further accumulation of lipid droplets under our experimental setting, because production of lipid droplets might favor virus replication during the first hours of infection, while lipolysis benefits virus replication at later time points [83]. The precise mechanistic basis for these effects on lipoprotein metabolism and lipid droplet biogenesis remains to be identified in additional studies.

SARS-CoV-2 infection has been associated with apoptotic cell death in various cell types, including liver cells. The mechanisms underlying apoptosis during SARS-CoV-2 infection are multifaceted and may involve the virus-induced activation of apoptotic pathways. Apoptosis will occur to eliminate the stressed cells if the ER stress is sustained, and cells fail to re-establish the proper ER protein folding capacity. Prolonged or severe ER stress can activate pro-apoptotic signaling pathways through the UPR [84]. Given the central role of PERK, ATF6, and IRE1α in UPR signaling, it is likely that these UPR mediators may play a critical role in ER stress-induced apoptosis. The roles of each UPR mediator and their possible links to intrinsic apoptotic pathways have been well described [85,86,87]. CHOP is a key mediator of ER stress-induced apoptosis. It regulates the expression of pro-apoptotic genes while suppressing anti-apoptotic factors [88]. Interestingly, at the transcriptional level, pre-existing ER stress prior to virus infection attenuated the induction of PERK mediating activation of downstream pro-apoptotic CHOP in transduced cells (Figure 5). This effect was more pronounced with THA than TUN, most probably due to disruption of intracellular Ca^2+^ homeostasis-mediated apoptosis [89]. Surprisingly, the reduction of anti-apoptotic BCL2 expression was attenuated by TUN but not THA, demonstrating again the differential effect of the two mechanistically different ER stressors. Furthermore, the incomplete rescue from apoptosis by TUN in transduced cells may be explained by insufficient alteration of cytosolic Ca^2+^ caused by short treatment of ER stressors or the contribution of factors other than just Ca^2+^ dysregulation that could negatively impact the early adaptive phase of apoptosis during ER stress response. Activation of JNK during the initial phase that precedes splicing of XBP1 has been shown to inhibit apoptosis in ER stress response [90]. Pre-existing ER stress with TUN but not THA indicated that IRE1α induced *XBP1* mRNA splicing could promote the expression of specific genes to alleviate ER stress in transduced cells (Figure 2 and Figure 5). Furthermore, JNK is known to directly inhibit BCL2 anti-apoptotic activity and high levels of BCL2 were indeed associated with reduced apoptosis in JNK knockdown cells [91]. In this study, no attempts were made to study the activation/cleavage of caspase-8, known as the hallmark of the extrinsic apoptotic pathway; therefore, our work focused only on the pivotal role of BCL2 in the initiation of the intrinsic pathway. Our results showed that pre-existing TUN-induced ER stress was able to suppress the activation of JNK-induced apoptosis in transduced cells, most likely through modulation of BCL2 expression or/and other BCL2 family proteins (Figure 5).

## 5. Conclusions

In conclusion, the use of two unrelated functionally different ER stress inducers in this study, rather than an attempt to use individual ER stressors, may hold promise in better understanding the complex interplay between acute pre-existing ER stress prior to infection and SARS-CoV-2 invasion into liver. While caution should be taken in interpreting results with different ER stress inducers that may have various effects beyond ER stress, this study has provided evidence showcasing the modulation of a short-term pre-existing ER stress response during SARS-Co-V2 infection. The consistent attenuation of virus entry into Huh7 cells has been associated with differential modulation of several interconnected critical metabolic control points in host cells leading to development of early cellular defense mechanisms capable of re-programming the UPR pathways, glucose/insulin signaling, lipid metabolism, and apoptosis (Figure 6). How UPR activation during an acute pre-existing ER stress could interfere with the entry of SARS-CoV-2 into host cells remains to be further elucidated in other target cells. Solving the detailed mechanisms of UPR modulating all these interconnected metabolic pathways will certainly help us to better understand the refined defense strategy against viral infection in stressed cells. The results discussed here may pave the way for several exciting future perspectives that could contribute to identification of new targets and development of effective therapeutic strategies for COVID-19.

## Figures and Tables

**Figure 1 cells-13-00769-f001:**
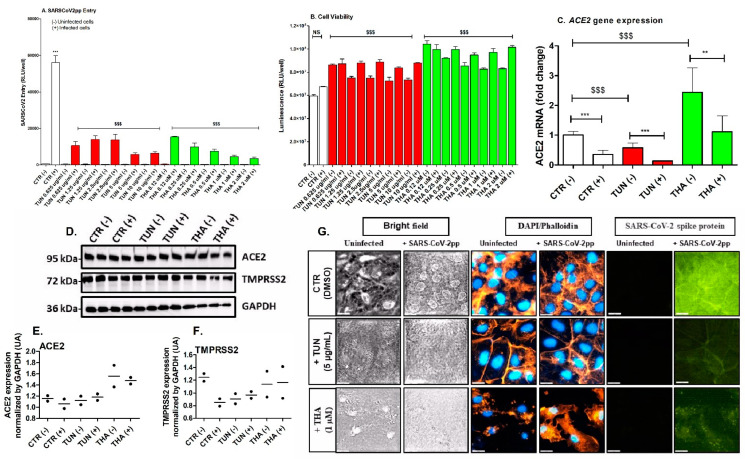
Pre-existing ER stress efficiently inhibits SARS-CoV-2 entry into Huh7 cells independently of ACE2 expression. Huh7 cells were pre-treated with increasing concentrations of Tunicamycin (TUN: 0.65 µg/mL–10 µg/mL) and Thapsigargin (THA: 0.12 µM–2 µM) for 6 h prior infection with SARS-CoV-2pp (35,000 RLU/mL) for 48 h. At 48 p.i. infected and uninfected cells (control) were used to quantify SARS-CO-V2pp entry using luciferase assay (Panel **A**) and to determine cell viability (Panel **B**). In separate experiment, cells were pre-treated with either TUN (5 µg/mL) or THA (5 µM) prior infection with SARS-CoV-2pp (35,000 RLU/mL). After 48 h p.i. cells pellet samples were used for ACE2 mRNA quantification by RT-PCR (Panel **C**) and proteins analysis by western blot (Panel **D**). Density of duplicate bands of ACE2 and TMPRSS2 proteins from panel D was quantified, normalized to housekeeping protein GAPDH and plotted as mean of two replicates (Panels **E**,**F**). In subsequent experiment, cells were treated with ER stress inducers prior SARS-CoV-2pp infection (48 h p.i.) and fixed and analyzed by immunofluorescence staining using Dapi for cells nuclei (blue), Phalloidin stain for F-actin (orange), and SARS-CoV-2 anti-spike protein antibodies (green). Images were taken and analyzed with resolution of fluorescence microscopy up to 20 µm (Panel **G**). Values are plotted as the mean ± SD. *p* values were calculated using a two-tailed Student *t*-test. ** *p* < 0.001 and *** *p* < 0.0001 (uninfected control cells vs. infected cells) and $$$ *p* < 0.0001 (effect of ER stress inducers compared to infected cells). NS, not significant.

**Figure 2 cells-13-00769-f002:**
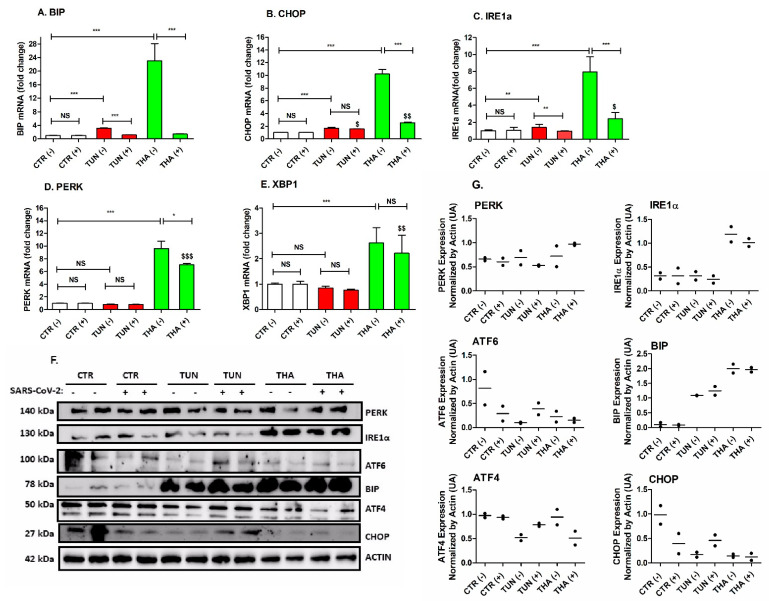
Pre-existing ER stress suppresses the activation of UPR gene markers in infected Huh7 cells. Cells were treated with either TUN (5 µg/mL) or THA (1 µM) for 6 h prior to infection with SARS-CoV-2pp (35,000 RLU/mL) for 48 h. Infected cells (+) and uninfected cells (−) were used for RNA extraction and total protein to measure the expression of various ER stress genes by qRT-PCR. Values are plotted as the mean ± SD (Panels **A**–**E**) and proteins by Western Blot (Panel **F**). Density of duplicate bands of target ER stress proteins was quantified, normalized to housekeeping protein actin and plotted as mean of two replicates (Panel **G**). *p* values were calculated using a two-tailed Student *t*-test. * *p* < 0.05, ** *p* < 0.001 and *** *p* < 0.0001 (compared to control cells). $ *p* < 0.05; $$ *p* < 0.001 and $$$ *p* < 0.0001 (effect of virus infection compared to control cells). NS, not significant.

**Figure 3 cells-13-00769-f003:**
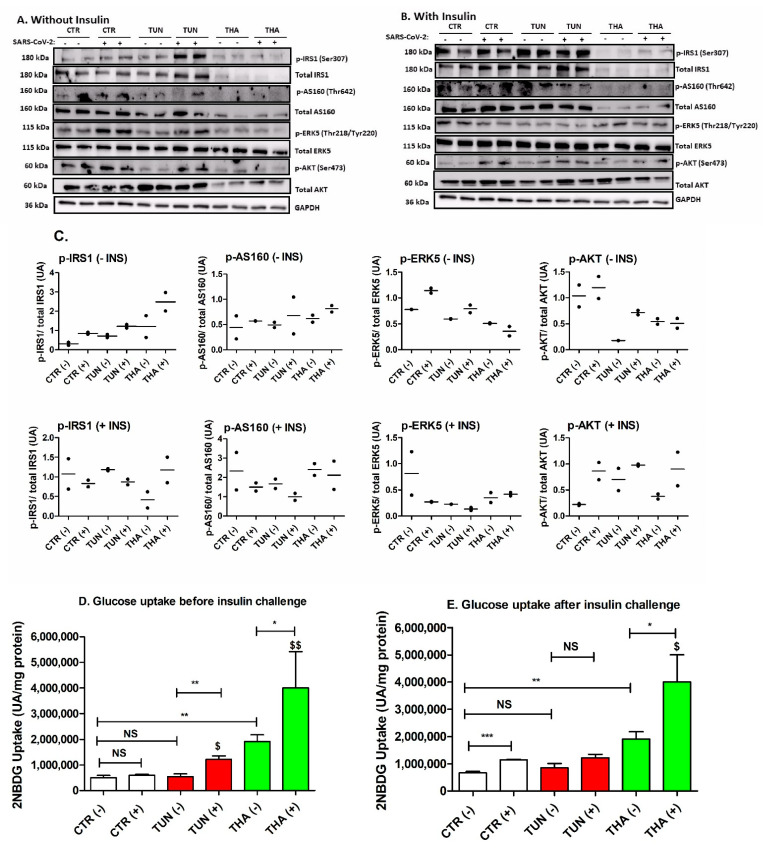
Pre-existing ER stress influences insulin signaling and glucose uptake in infected Huh7 cells. ER stress was induced by TUN and THA for 6h prior SARS-CoV-2pp infection (35,000 RLU/mL) then cells were supplemented with or without bovine insulin (250 nM) for a 15 min acute challenge. Cell pellets were lysed and subjected to western blot analysis to determine the expression of downstream insulin pathway protein before (Panel **A**) and after acute insulin challenge (Panel **B**). Bands density for corresponding target proteins from panels A and B were quantified and plotted as mean of two replicates (Panel **C**). In subsequent experiments, cells were used to measure 2NBD glucose uptake before (Panel **D**) and after insulin challenge (Panel **E**). Values are plotted as the mean ± SD. *p* values were calculated using a two-tailed Student *t*-test. * *p* < 0.05, ** *p* < 0.001 and *** *p* < 0.0001 (compared to control cells). $ *p* < 0.05 and $$ *p* < 0.001 (Effect of virus infection compared to control cells). NS, not significant.

**Figure 4 cells-13-00769-f004:**
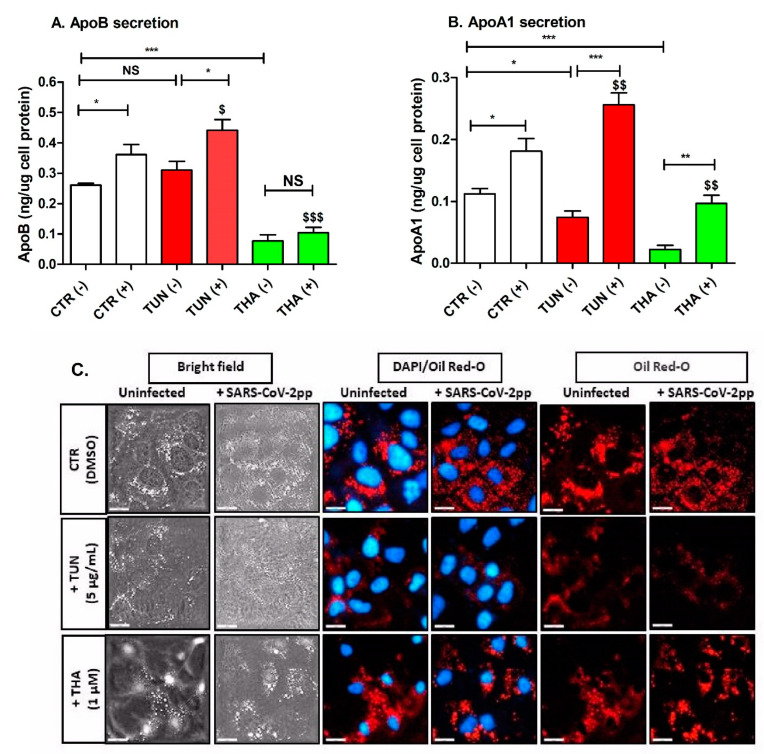
TUN and THA differentially modulate intracellular accumulation of lipid droplets and lipoproteins secretion by SARS-CoV-2 infected Huh7 cells. Cells were treated with TUN and THA for 6h prior SARS-CoV-2pp infection (35,000 RLU/mL). At 48 h p.i. collected media from infected cells and uninfected cells (control) was used to quantify levels of secreted apoB (Panel **A**) and apoA1 (Panel **B**) by ELISA. In a subsequent experiment, cells were treated with ER stress inducers prior SARS-CoV-2pp infection (48 h p.i.), fixed and analyzed by immunofluorescence staining using Dapi for cells nuclei (blue) and Oil Red-O for lipid droplets staining (red). Images were taken and analyzed with resolution of fluorescence microscopy up to 20 µm (Panel **C**). Values are plotted as the mean ± SD. *p* values were calculated using a two-tailed Student *t*-test. * *p* < 0.05, ** *p* < 0.001, and *** *p* < 0.0001 (compared to control cells). $ *p* < 0.05, $$ *p* < 0.001, $$$ *p* < 0.0001 (effect of virus infection compared to control cells). NS, not significant.

**Figure 5 cells-13-00769-f005:**
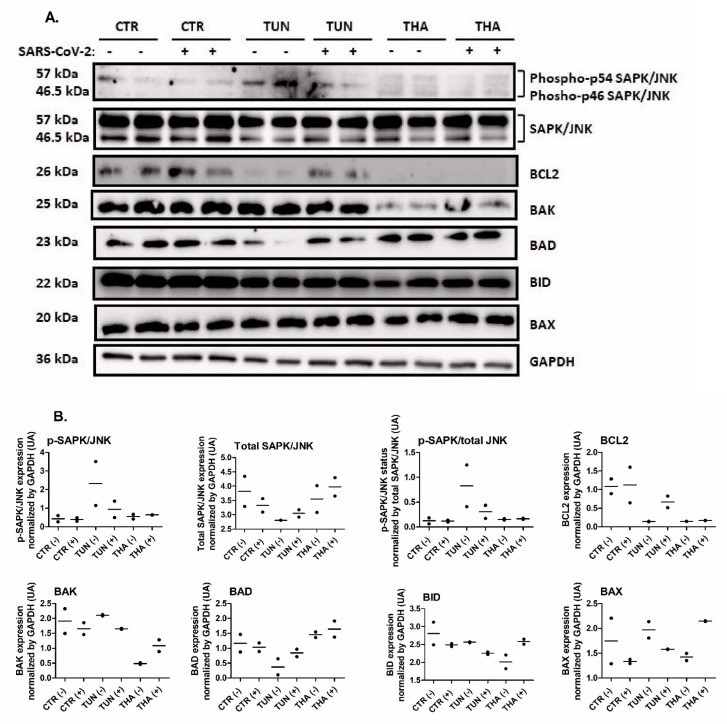
Pre-existing ER stress alters stress-activated protein kinase (SAPK/JNK) signaling and apoptosis pathway. ER stress was induced by TUN (5 µg/mL) and THA (1 µM) for 6h prior SARS-CoV-2pp infection (35,000 RLU/mL). At 48 h p.i. cell pellets we lysed and subjected to western blot analysis to determine the expression of phospho-SAPK/JNK, pro-apoptotic and anti-apoptotic proteins (Panel **A**). Immunoblot protein bands were quantified and plotted as density mean values of two replicates (arbitrary units, UIA) of corresponding (Panel **B**).

**Figure 6 cells-13-00769-f006:**
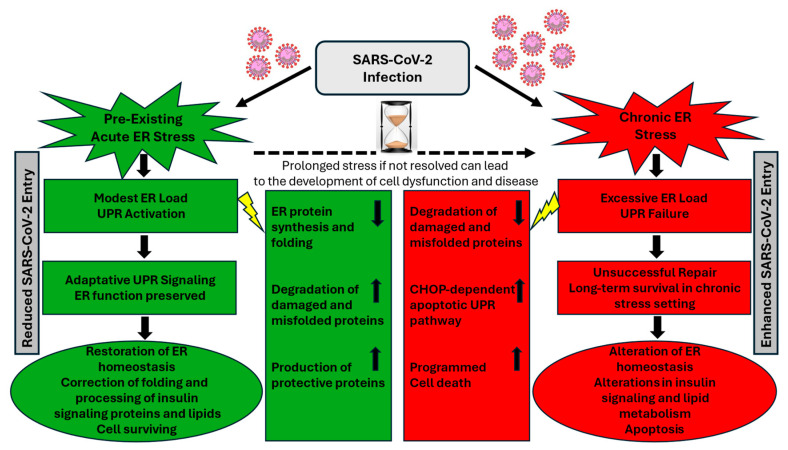
The status of cellular ER stress may play a crucial role in determining whether stressed cells will survive SARS-CoV-2 infection or undergo programmed cell death. The transition from pre-existing acute stress to chronic stress following SARS-CoV-2 infection is a complex process that can vary depending on specific circumstances such as the nature of stressors, duration of stress, and the capacity of infected host cells to cope with the stress. Activation of UPR during a pre-existing ER stress may represent an early host defense mechanism to restore normal cell function and ER homeostasis by promoting the correct folding and processing of proteins and lipids, ultimately leading to reduced SARS-CoV-2 entry into host cells. If the stress is not resolved quickly and remains persistent and prolonged due to UPR hyperactivity and inadequate cellular repair mechanisms, stressed cells become more prone to virus infection, which can cause an aggravated pathological chronic ER stress-associated cellular dysfunction, ultimately affecting the clustering and localization of viral entry receptors on the cell surface, impairing SARS-Co-V-2 entry into host cells.

## Data Availability

The data presented in this study are available in article.

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
