# Peer review of "Thapsigargin and Tunicamycin Block SARS-CoV-2 Entry into Host Cells via Differential Modulation of Unfolded Protein Response (UPR), AKT Signaling, and Apoptosis"

_cells, 2024, doi:10.3390/cells13090769_

Round 1
Reviewer 1 Report
Comments and Suggestions for Authors
With the work entitled “Thapsigargin and Tunicamycin block Sars-CoV-2 entry into host cells via differential modulation of unfolded protein response (UPR), AKT signaling and Apoptosis” the authors described some signaling pathways modulated during virus infection. Through various molecular tests, they were able to reach certain conclusions that led to positively testing their hypotheses. These data are important for the initial biological knowledge of the physiological dynamics of the virus and its pathogenicity, opening perspectives on possible developments of medicines used in the future if necessary. The professional language in English is consistent, as are the results presented through graphs and images. The hypothesis discussed is in accordance with the data presented and the literature provided at the end of the manuscript.
I have some suggestions and queestions.
1. The Introduction could provide more information about the signaling pathways studied in other infectious models.
2. At the end of the work, it is suggested that the authors try to make a schematic illustration to better disseminate the results.
3. TUN and THA are molecules that interact with numerous cellular targets, not only related to ER stress. The authors could discuss this observation as a self-criticism.
4. As a measure to provide unexpected results with the use of inhibitory/activating molecules, did the authors consider using gene silencing to test some of the hypotheses, such as inhibition of the ACE2 receptor?
5. Why was the expression of Interferon I and II not analyzed, given that they are also induced by UPR?
Reviewer 2 Report
Comments and Suggestions for Authors
Thapsigargin and Tunicamycin block SARS-CoV-2 entry into host cells via differential modulation of unfolded protein response (UPR), AKT signaling and Apoptosis
By Al Otaibi et al.
The study attempts to shed light on the interplay between SARS-Co-V2 infection and the unfolded protein response (UPR) signaling, given that patients with altered UPR seem to be more prone to severe COVID19. To mimic their conditions, the Authors treated HUH-7 cells with Tunicamycin (TUN) and Thapsigargin (THA) to induce ER stress. Then, pseudotyped particle (SARS-CoV-2pp) entry into host cells was measured using by a luciferase assay, while mRNA and protein expression were evaluated by RT-qPCR and Western Blot.
They found that TUN and THA inhibited the entry of SARS-CoV-2pp into host cells at non-toxic concentrations. A lower entry level was not associated with modulation of ACE2 expression.
Both TUN and THA significantly reduced the expression of stress-inducible ER chaperone GRP78/BiP in infected cells. In contrast, the IRE1-XBP1s and PERK-eIF2α-ATF4-CHOP signaling pathways were downregulated with THA treatment but not TUN in infected cells.
Insulin-mediated glucose uptake, phosphorylation of Ser307 IRS-1 and downstream p-AKT were enhanced with THA in infected cells. TUN and THA also influenced lipid metabolism and apoptosis.
These findings suggest that short-term pre-existing ER stress prior to virus infection induces a specific UPR response in host cells capable to counteract the stress-inducible signaling, thereby depriving SARS-Co-V2 of essential components for their entry and replication.
English: Although I would still recommend the manuscript to be read by a native English speaker, the language is good.
MAJOR COMMENTS
1) The Authors should describe in detail the PV used, at least in brief, in M&M and in the text, at their first mention. Are they lentiviral? HIV-1-based? VSV based? What Spike are they using? Do the PV carry any gene? In addition, it is important to avoid calling the PV SARS-CoV-2 (e.g. line 240).
2 The Authors keep mentioning infection, but treatment of cells with pseudotyped viruses (PV) is called transduction.
3 Figure 1. This is a crucial figure in the paper, but many controls are missing.
• The most important is a PV pseudotyped with another viral glycoprotein, typically VSV-G; this control is missing everywhere and it is crucial.
• there is no quantification of ACE-2 western blot to show a reduction in ACE2 espression. It would be more interesting to include an ACE-2 and TMPRSS staining in Fig 1E.
• What is the last panel in Fig. 1E? There is no fluorescence, only background staining. I believe that claiming one can stain with anti SARSCOV2 spike antibody after transduction with S-PV should be proven and not taken for granted.
• No quantification in levels of fluorescence and in quite a few other figures.
• No statistics in Figs 2G or 3C, where very small differences between treatments may be due to random experimental issues.
• No attempts to confirm the data in other cell lines, such as Calu-3 or Vero-TMPRSS, cells that are more often used that HUH7.
For these reasons, I believe that the paper does not tackle the question appropriately and cannot be published in its present form.
